Subject Area:
molecular biology/systems biology

Keywords:
arachidonic acid, asthenozoospermia, mitogen-activated protein kinases, metabolomics, seminal plasma

Authors for correspondence:
Xuejun Shang
e-mail: shangxj98@sina.com
Bingfang He
e-mail: bingfanghe@njtech.edu.cn
Qi Zhang
e-mail: nancyzhang03@hotmail.com

# Abnormal arachidonic acid metabolic network may reduce sperm motility via P38 MAPK

Lisha Yu[1], Xiaojing Yang[1], Bo Ma[1], Hanjie Ying[2], Xuejun Shang[3], Bingfang He[1] and Qi Zhang[1]

[1]College of Pharmaceutical Sciences, Nanjing Tech University, Nanjing 210009, People's Republic of China
[2]College of Biotechnology and Pharmaceutical Engineering, Nanjing Tech University, Nanjing 211816, People's Republic of China
[3]Department of Andrology, Jinling Hospital, School of Medicine, Nanjing University, Nanjing 210002, People's Republic of China

HY, 0000-0002-4061-0001; QZ, 0000-0002-5292-9087

Asthenozoospermia is a common cause of male infertility, the aetiology of which remains unclear in 50–60% of cases. The current study aimed to characterize metabolic alterations in asthenozoospermic seminal plasma and to explore the signalling pathways involved in sperm motility regulation. At first, high-performance liquid chromatography–electrospray ionization–tandem mass spectrometry was used to detect the targeted metabolic network of arachidonic acid (AA). Metabolomic multivariate data analysis showed significant distinction of AA metabolites between asthenozoospermic and healthy seminal plasma. AA as well as its lipoxygenase (LOX) and cytochrome P450 metabolites were found to be abnormally increased, while cyclooxygenase (COX) metabolites were complicatedly disturbed in asthenozoospermic volunteers compared with those in healthy ones. *In vitro* experiments and western blot analysis of sperm cells revealed a decrease in sperm motility and upregulation of sperm phosphor-P38 induced by AA. P38 inhibitor could increase AA-reduced sperm motility. Also, all the inhibitors of the three metabolic pathways of AA could block AA-induced P38 mitogen-activated protein kinase (MAPK) activation and further improve sperm motility. We report here for the first time that an abnormal AA metabolic network could reduce sperm motility via P38 MAPK activation through the LOX, cytochrome P450 and COX metabolic pathways, which might be an underlying pathomechanism of asthenozoospermia.

## 1. Introduction

Asthenozoospermia, characterized by decreased progressive motility or an absence of sperm motility in fresh ejaculate, is detected in more than 50% of infertile males [1,2]. Unfortunately, no definitive causes can be identified in 50–60% of clinical cases, leading to the lack of specific and effective treatment of asthenozoospermia. Therefore, further investigations into the pathological mechanisms of asthenozoospermia are urgently required for clinical diagnosis and therapy [3]. The detection of semen parameters is one of the diagnostic approaches to asthenozoospermia, but currently semen examination only includes seminal volume, pH, sperm concentration, motility and morphology, as recommended by the World Health Organization (WHO). The information obtained from semen by biochemical analysis is, unfortunately, very limited. Considering seminal plasma, the survival microenvironment for sperm not only provides energy and nutrients for sperm but also plays an essential role in sperm capacitation, fertilization and other biological functions [4]. It will be of great significance to investigate seminal plasma metabolites and further explore any correlations between metabolic disorders and sperm lesions.

Metabolomics, the study of the complete repertoire of small molecules in cells, tissues, organs and biological fluids, represents a major and rapidly evolving aspect of biology [5–7]. The metabolomic analytical platforms (nuclear magnetic resonance (NMR), gas chromatography–mass spectrometry (GC-MS), liquid chromatography–tandem mass spectrometry (LC-MS/MS), etc.) are capable of accurately measuring hundreds or thousands of small molecules in biological samples [8–10]. Recently, the strategy for metabolomics moved on from the discovery of changed metabolites into the confirmation of biomarkers, and even further the analysis of metabolic pathways and the investigation of relevant molecular mechanisms [11,12].

However, so far, only Gilany et al. [13], Jafarzadeh et al. [14] and Zhang et al. [15] have reported on the metabolomic analysis of seminal plasma from clinical patients. They all adopted a non-targeted method (Raman spectroscopy or $^1$H NMR), thus only a few compounds were identified and the metabolite profiles obtained provided very limited information. In addition, none of these authors explained the correlation between disease and metabolites, let alone the molecular mechanism of asthenozoospermia.

According to Am-In et al. [16], Lu et al. [17] and Kim et al. [18], abnormal lipid metabolism may contribute to male infertility. Arachidonic acid (AA, 20:4n-6), which plays a significant role in lipid metabolism, was found to have a concentration-dependent inhibitory effect on the motility of human sperm in in vitro experiments as early as 1986 [19], whereas Andersen et al. [20] reported that AA levels correlated positively between serum and spermatozoa, and sperm motility was positively related to spermatozoa AA level. It was quite confusing that the findings mentioned above about the association between AA level and sperm motility seemed conflicting. Apparently, researchers have come to realize the impact of AA on sperm function, but research in this area was relatively fragmented and somehow researchers failed to reach a consensus about the exact role of AA in sperm function. Therefore, we used a combination in vivo and in vitro study to systematically explain the association between AA and sperm motility.

It is known that the AA metabolic network includes three pathways: cyclooxygenase (COX), lipoxygenase (LOX) and cytochrome P450 (CYP450), which are involved in signal transduction pathways in several biological processes [21]. The metabolites of AA, including prostaglandins (PGs), leukotrienes, hydroperoxyeicosatetraenoic acids (HpETEs) and epoxyeicosatrienoic acids (EETs), are bioactive factors which exert biological effects in vivo [22]. Fiebich et al. [23] found that prostaglandin $E_2$ (PGE$_2$) stimulated interleukin 6 release in U373 MG human astroglioma cells via the activation of P38 MAPK. Lee et al. [24] found that the activation of MAPKs was responsible for PGE$_2$ secretion in human tracheal smooth muscle cells. In addition, Evans et al. [25] reported that AA could induce brain endothelial cell apoptosis via P38 MAPK.

Three categories of MAPK families have been well characterized: extracellular signal-regulated kinases (ERK1/2), c-Jun N-terminal kinase (JNK) and P38 MAPK. According to Almog et al. [26–28], the MAPK signalling pathway plays a significant role in spermatogenesis, sperm meiosis, capacitation and acrosome reaction. Liu et al. [29] reported that the MAPK cascade is required for sperm activation in Caenorhabditis elegans. Thus, it was reasonable to speculate that an abnormal AA metabolic network in seminal plasma might affect sperm motility via the MAPK pathway.

In our study, high-performance liquid chromatography/electrospray ionization tandem mass spectrometry (HPLC-ESI-MS/MS) was performed for the targeted analysis of AA and its metabolites to elucidate the correlation between AA metabolism and asthenozoospermia. Subsequently, to clarify how the AA metabolic network affected sperm motility, protein levels of the MAPK pathway were examined after AA treatment by in vitro experiments and western blot analysis. The impacts of the AA metabolic pathways (COX, LOX and CYP450) on the MAPK pathway and sperm motility were finally evaluated by using specific inhibitors. This study aimed to explore whether and how the AA metabolic network regulated sperm motility via the MAPK signalling pathway.

# 2. Material and methods

## 2.1. Subjects

Human semen samples included in this study were donated by volunteers, aged 24–50 years, from Jinling Hospital, Nanjing, China. All procedures carried out in relation to the present study were in accordance with the Declaration of Helsinki and were approved by the Ethics Committee of Nanjing Tech University and Jinling Hospital (Nanjing, China). Approval was obtained on 5 March 2013 (approval no: 2013GJJ-078). Written informed consent forms were obtained from each subject. The routine parameters acquired from all subjects, including age, body mass index (BMI), sperm motility indexes and duration of abstinence, are shown in electronic supplementary material, table S1.

## 2.2. Materials

Alkane series (C8–C40), AA, Percoll and anisomycin were obtained from Sigma-Aldrich (St. Louis, MO, USA). HPLC-grade methanol was purchased from Merck (Darmstadt, Germany). Biggers–Whitten–Whittingham (BWW) medium was purchased from Genmed Scientifics (Wilmington, DE, USA). Radioimmuno-precipitation assay (RIPA) protein standard lysis buffer and phenylmethylsulfonyl fluoride (PMSF) were obtained from Beyotime Institute of Biotechnology (Jiangsu, China). Anti-P38 MAPK, anti-phosphor-P38 MAPK, anti-ERK, anti-phosphor-JNK, anti-JNK, anti-phosphor-ERK, anti-GAPDH, SB203580 and secondary antibodies were obtained from Cell Signaling Technology (Danvers, MA, USA). Diclofenac sodium was purchased from China Pharmaceutical Biological Products Analysis Institute (Beijing, China). SC-560 and NS-398 were obtained from Cayman Chemical (Ann Arbor, MI, USA). C26 was obtained from Toronto Research Chemicals (Toronto, Ontario, Canada). AA-861 was obtained from Enzo Life Sciences (New York, NY, USA).

## 2.3. Sample collection and pre-preparation for metabolomic study

Semen samples were obtained by masturbation after 3–5 days of ejaculatory abstinence from volunteers. After semen liquefaction and analysis of basic parameters, samples were divided according to the WHO Laboratory Manual for the Examination and Processing of Human Semen [30] into a healthy group (n = 33, sperm progressive motility greater than 50%) and an

royalsocietypublishing.org/journal/rsob   Open Biol. 9: 180091

asthenozoospermic group ($n = 30$, sperm progressive motility less than 50%). For the metabolomic study, liquefied semen samples were centrifuged at 4000 rpm, 4°C, 10 min, to separate spermatozoa from the seminal plasma, followed by a second centrifugation step performed at 14 000 rpm, 4°C, for cellular debris removal. The seminal plasma obtained was stored at −80°C in aliquots. Another batch of semen samples with normal parameters in accordance with WHO criteria were collected for molecular mechanism research.

## 2.4. HPLC-ESI-MS/MS method

A total of 200 µl of seminal plasma was spiked with 50 µl of IS (AA-d$_8$, 200 ng ml$^{-1}$) for 30 s and vortex mixed with 750 µl methyl tertiary-butyl ether for 10 min. After centrifugation at 12 000 rpm for 10 min at 4°C, 700 µl of supernatant was evaporated to dryness under vacuum in a speedvac concentrator. The residue was reconstituted in 100 µl acetonitrile/water (50 : 50, v/v) and mixed for 10 min. Then, 8 µl of the final supernatant (80 µl) after centrifugation was injected for HPLC-ESI-MS/MS analysis.

HPLC-ESI-MS/MS analysis was carried out on a Shimadzu HPLC-20A system (Shimadzu Corporation, Japan) and an API 4000 Q-trap MS/MS system (AB Sciex, USA) equipped with a turbo ion spray inlet in negative-ion electrospray mode. Samples were separated on a Kromasil C18 column (150 × 2.1 mm, 5 µm) with the column temperature at 35°C. The mobile phase consisted of 0.1% (v/v) acetic acid in water (A) and acetonitrile containing 0.1% (v/v) acetic acid (B). The flow rate was set at 0.3 ml min$^{-1}$. The gradient programme was as follows: 0.01–15 min, from 30% B to 60% B; 15–25 min, linear gradient to 80% B; 25–36 min, linear gradient to 90% B; 36–36.5 min, linear gradient to 30% B; 36.5–52 min, hold at 30% B to equilibrate the column.

The ESI source was set in negative ionization mode and multiple reaction monitoring (MRM) at unit resolution was employed to monitor the transitions of the molecular ions for compounds. The turbo gas temperature was set at 550°C. The pressures of the auxiliary gas and nebulizing gas were both set at 55 kPa. The ion spray voltage was set at −4500 kV. The specific precursors and product ions of the standards we used to confirm AA metabolites in human seminal plasma are shown in electronic supplementary material, table S2. Data acquisition was performed with Analyst 1.5.1 software.

The automatic detection of the integral peak width was set to 1 s and peaks with a signal to noise ratio higher than 10 were defined as a minimum threshold for quantification. A list of intensities of the peaks detected was then generated for the first chromatogram, using the $Rt$-$m/z$ data pairs as identifiers.

## 2.5. Data process and multivariate data analysis

Each peak area was normalized using IS (AA-d8) before data analysis. Univariate statistical analysis was conducted by SPSS 16.0 software using Student's $t$-test analysis. A $p < 0.05$ was considered statistically significant. Metabolomic multivariate data analysis was carried out using SIMCA-P 11.0 for principal component analysis (PCA) and partial least-squares discriminant analysis (PLS-DA), as well as MetaboAnalyst 3.0 for assessment of the variable influence on projection (VIP) value, fold change and clustering heatmap analysis. Score plots of unsupervised PCA and supervised PLS-DA exhibiting similarities and differences between groups were obtained by visualizing sample clustering and segregation.

Metabolites with a larger VIP value had a greater contribution to group separation. Differential metabolites were determined according to $p$-value ($p < 0.05$) and VIP (VIP > 1.1).

## 2.6. Quality control and quantitative analysis of differential metabolites

Quality control (QC) was performed to investigate instrument precision after an interval of eight seminal plasma samples. According to §2.3, the peak area of IS in each QC sample was obtained. The precision was expressed as the relative standard deviation (RSD).

Representative compounds of different AA metabolites were selected to carry out the validation of the current method. Referring to Zhu et al. [31] and Amoako et al. [32], standards of hydroxyeicosatetraenoic acids (HETEs), EETs and hydroxy/hydroperoxyeicosatrienoic acids (DHETs) were diluted as follows: 0.01, 0.05, 0.1, 0.5, 1, 5, 25, 50 ng ml$^{-1}$. Also, standards of AA and PGs were diluted to 0.5, 1, 2.5, 10, 25, 100, 200, 400 ng ml$^{-1}$. Each sample was divided into two parts, one for analysis of HETEs, EETs and DHETs, and the other for detection of AA and PGs. Samples for AA and PG analysis were diluted 200-fold before preparation, as described in §2.3, whereas those for other analytes were prepared as described in §2.3. Linear least-squares regression analysis was conducted employing a weighting factor of the reciprocal of the concentration squared ($1/X^2$). The correlation coefficients ($R^2$) were obtained using the ratio of peak area of each analyte to that of IS in the analysed concentration range.

Precision and accuracy were determined by using six samples per concentration at three different levels (HETEs, EETs and DHETs: 0.2, 2.5 and 20 ng ml$^{-1}$, AA and PGs: 1, 50, 200 ng ml$^{-1}$). Samples were prepared as described in §2.4. Intra-day precision and accuracy were evaluated on the same day, while inter-day precision and accuracy were evaluated over 3 days. Precision was assessed and the values were expressed as the RSD. Accuracy was calculated as the percentage bias from the nominal concentration (% bias).

## 2.7. Human spermatozoa preparation for *in vitro* experiments

Semen samples with normal parameters in accordance with WHO criteria were collected. After liquefaction in a water bath maintained at 37°C, semen from several subjects was combined as a pool of healthy sample. Semen samples were then washed with BWW medium and centrifuged using discontinuous Percoll gradients [33] at 300$g$ for 30 min until no non-sperm cells were observed under phase contrast microscopy. After that, the sperm cells were diluted to 10 × 10$^6$/ml in BWW and incubated for 1 h at 37°C under humidified air with 5% CO$_2$ for preincubation.

## 2.8. Spermatozoa treatment to evaluate the impact of AA on the MAPK pathway and sperm motility

After preincubation, human spermatozoa with normal motility (a + b > 50%) were treated with AA (0 µM, 20 µM, 40 µM, 60 µM) for 3 h. The percentage of sperm motility in each group was evaluated by computer-aided sperm

analysis (CASA). Data were expressed as the mean ± s.d. and analysed by Student's *t*-test, with $p < 0.05$ set as the level of statistical significance. Sperm proteins were extracted and expressions of phosphor-P38, P38, phosphor-ERK1/2, ERK1/2, phosphor-JNK and JNK were detected by western blot.

According to the AA-induced protein level change, specific MAPK inhibitors were added (P38 inhibitor SB203580, 50 μM) to spermatozoa for 20 min before AA (60 μM) treatment for 3 h. Healthy human sperm without AA administration was set as the blank control, while sperm incubated with AA (60 μM) and anisomycin (P38 MAPK activator, 25 μg ml$^{-1}$), respectively, were set as positive controls.

## 2.9. Spermatozoa treatment to investigate the influence of AA metabolic pathways on MAPKs and sperm motility

Selective COX-1 inhibitor SC-560 (5 μM), selective COX-2 inhibitor NS-398 (20 μM), non-selective COX inhibitor diclofenac sodium (50 μM), 5-LOX inhibitor AA-861 (10 μM) and CYP2J2 inhibitor C26 (10 μM) were used to incubate spermatozoa before AA administration. Sperm motility and MAPK expression were evaluated to discover whether such inhibitors could block AA-induced MAPK activation and sperm defects. Healthy human sperm was used as the blank control, while sperm incubated with AA (60 μM) was set as the positive control.

## 2.10. Western blot analysis

After being treated as mentioned in §§2.8 and 2.9, sperm cells were washed twice with phosphate-buffered saline and then incubated in RIPA lysis buffer containing 1 mM PMSF and protease inhibitor mixtures for 10 min on ice, according to the manufacturer's instructions (Beyotime Institute of Biotechnology, Jiangsu, China). After centrifugation at 12 000 rpm for 10 min at 4°C, the supernatants were collected and assessed for protein concentration by the bicinchoninic acid method. Aliquots from each sample (30 μg) were separated on 12% sodium dodecyl sulfate–polyacrylamide gel electrophoresis and transferred onto a polyvinylidene difluoride membrane, and were then blocked by 5% non-fat milk for 1 h and incubated with the primary antibodies at 4°C overnight. All bands were visualized using an enhanced chemiluminescence system (Thermo Scientific, USA) after reacting with horseradish peroxidase-conjugated secondary antibodies for 1 h. The intensity was quantified using a ChemiScope 3400 Mini (Clinx Science Instruments, China). All samples were analysed in triplicate.

# 3. Results

## 3.1. Metabolic alterations involved in the AA metabolic network of seminal plasma based on HPLC-ESI-MS/MS

AA and 28 metabolites of AA listed in table 1 were detected in seminal plasma (chromatogram shown in electronic supplementary material, figure S1), including PGs (metabolized by COX), HETEs (metabolized by LOX and CYP450) and

**Table 1.** Arachidonic acid metabolites detected in human seminal plasma by HPLC-ESI-MS/MS.

| compound | m/z | Rt (min) |
|---|---|---|
| 13,14-dihydro-15-ketoPGE$_2$ | 351.000/235.000 | 10.14 |
| 8-isoPGF$_{2\alpha}$ | 353.000/193.000 | 10.21 |
| 13,14-dihydro-15-ketoPGF$_{2\alpha}$ | 353.000/291.000 | 10.42 |
| PGF$_{2\alpha}$ | 353.000/291.000 | 10.47 |
| 11β-PGF$_{2\alpha}$ | 353.000/309.000 | 10.92 |
| PGB$_2$ | 333.200/234.800 | 11.26 |
| 13,14-dihydro-15-ketoPGD$_2$ | 351.000/333.000 | 11.29 |
| PGD$_2$ | 351.000/233.000 | 11.20 |
| PGE$_2$ | 351.300/315.300 | 12.21 |
| PGJ$_2$ | 333.000/189.000 | 12.23 |
| PGF$_{2\beta}$ | 353.000/335.000 | 12.30 |
| 13,14-dihydro-15-ketoPGF$_{1\alpha}$ | 353.000/113.000 | 12.73 |
| 2,3-dinor TXB$_2$ | 341.000/123.000 | 13.79 |
| 15(S)-HpETE | 335.200/113.000 | 13.83 |
| 5(S)-HpETE | 335.000/317.000 | 15.26 |
| 14,15-DHET | 337.400/ 207.100 | 23.18 |
| 11,12-DHET | 337.400/167.000 | 26.77 |
| 20-HETE | 319.000/301.000 | 26.85 |
| 14,15-EET | 319.000/219.000 | 27.05 |
| 15-HETE | 319.000/219.000 | 27.85 |
| 8,9-EET | 319.000/123.000 | 27.86 |
| 11,12-EET | 319.000/167.000 | 28.09 |
| 11-HETE | 319.000/167.000 | 28.19 |
| 12-HETE | 319.000/179.000 | 28.34 |
| 5-HETE | 319.000/115.000 | 29.18 |
| tetranor-PGEM | 327.000/309.000 | 34.69 |
| 13(S)-HpODE | 311.000/113.000 | 36.69 |
| arachidonic acid | 303.000/259.000 | 37.59 |
| tetranor-PGFM | 329.000/311.000 | 39.27 |

EETs (metabolized by CYP450). Both PCA (figure 1*a*) and PLS-DA (figure 1*b*) score plots obviously separated the asthenozoospermic group from the healthy group, confirming that there are significant semen biochemical differences between the two groups. In the PCA model, $R^2X$ and $Q^2Y$ of four components were 0.723 and 0.781, respectively. A stronger discriminating power was obtained by the PLS-DA model. Parameters of $R^2X$, $R^2Y$ and $Q^2$ were used for internal cross-validation to test the reliability of the HPLC-ESI-MS/MS analysis, representing the explanation, fitness and prediction power, respectively. We adopted the leave-one-out cross-validation model, which divided samples into seven parts. One part of the sample was predicted using the model based on the remaining six parts. This process was repeated until all seven parts were predicted [34]. Two principal components (PC1 and PC2) were calculated with the $R^2X$, $R^2Y$ and $Q^2$ parameters of 0.504, 0.874 and 0.835, respectively. That is to say, two principal components could explain 50.4% of the LC-MS response variables and 87.4% of the sample variables and predict 83.5% of the sample

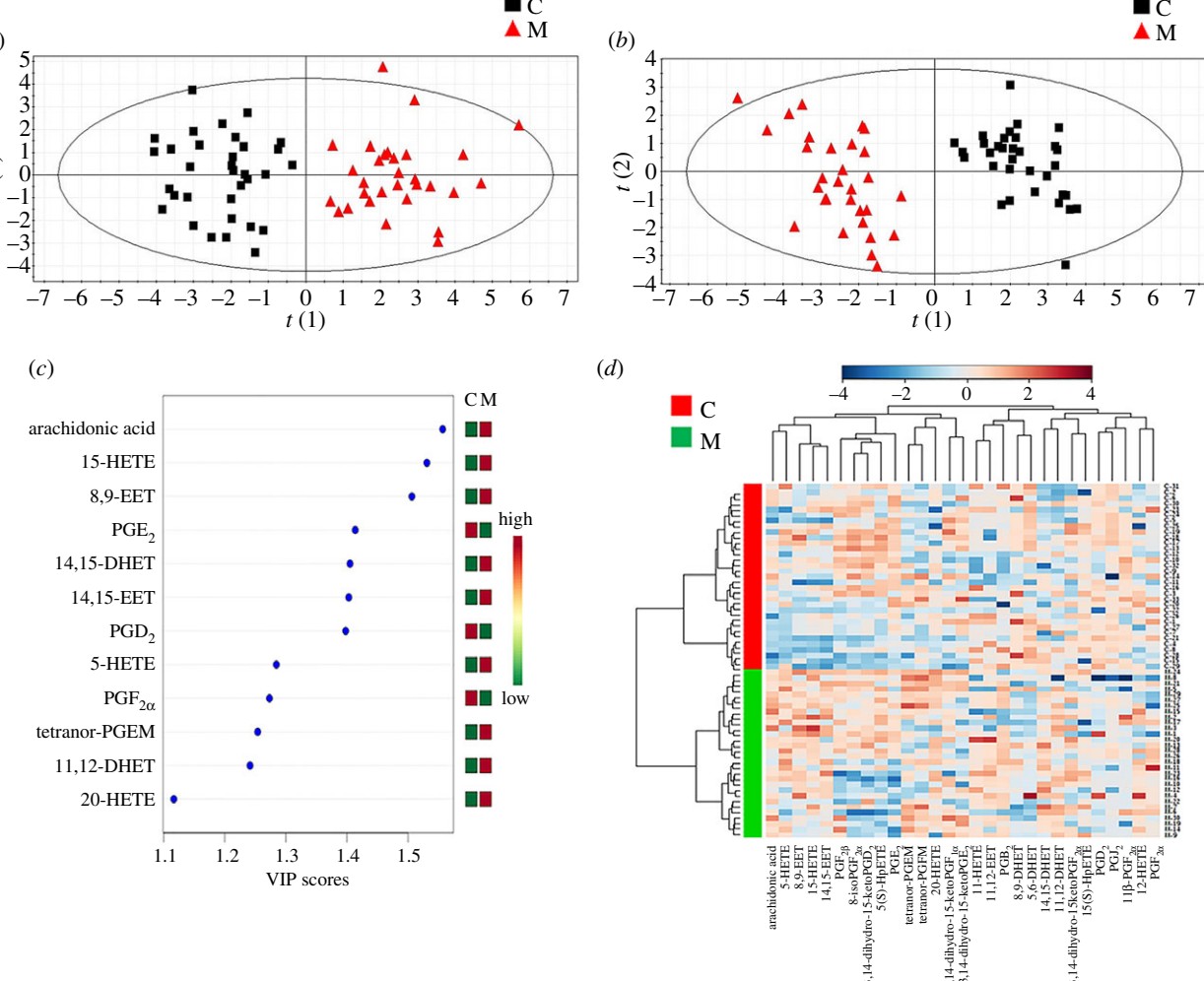

**Figure 1.** Metabolic profiling of AA metabolic alterations in asthenozoospermia patients. (*a*) PCA score plot of healthy and asthenozoospermic group (square, healthy group; triangle, asthenozoospermic group). (*b*) PLS-DA score plot of healthy and asthenozoospermic group (square, healthy group; triangle, asthenozoospermic group). (*c*) Top 12 significantly changed metabolites based on VIP scores of PLS-DA. (*d*) Heatmap of hierarchical clustering based on differential metabolites of importance for healthy and asthenozoospermic seminal plasma. (C refers to healthy group, M refers to asthenozoospermic group. Rows: samples; columns: differential metabolites; colour key indicates metabolite expression value, blue: low concentration, red: high concentration.)

variables, indicating that the model we established had good predictability for asthenozoospermia.

Hierarchical clustering analysis was performed with results shown in Heatmap (figure 1*d*), and distinct segregation was observed between asthenozoospermic and healthy groups. Based on VIP (VIP > 1.1), the top 12 significantly changed endogenous metabolites are listed in figure 1*c*. In order to better characterize the 12 metabolites, the VIP value, fold change and *p*-value of each analyte are listed in table 2.

AA exhibited higher contents (1.2-fold) in asthenozoospermic volunteers than in healthy volunteers and so did some of its metabolites: 5-HETE (1.5-fold), 15-HETE (1.5-fold) (metabolized by LOX) and 20-HETE (1.5-fold), 14,15-EET (1.3-fold), 8,9-EET (1.5-fold), 11,12-DHET (1.5-fold), 14,15-DHET (1.4-fold) (metabolized by CYP450), tetranor-PGEM (1.4-fold) (metabolized by COX), while the other compounds metabolized by COX, including $PGE_2$ ($-1.5$-fold), $PGD_2$ ($-1.2$-fold) and $PGF_{2\alpha}$ ($-1.3$-fold), decreased in the asthenozoospermic group, suggesting that an abnormal AA metabolic network might contribute to asthenozoospermia.

**Table 2.** Fold changes and *p*-value of significantly changed metabolites based on VIP in human seminal plasma analysed by HPLC-ESI-MS/MS.

| compound | *p*-value | fold change | VIP |
|---|---|---|---|
| arachidonic acid | 0.00001765 | 1.209 | 1.56 |
| 15-HETE | 0.00011173 | 1.501 | 1.54 |
| 8,9-EET | 0.00025715 | 1.481 | 1.52 |
| $PGE_2$ | 0.00010731 | $-1.545$ | 1.42 |
| 14,15-DHET | 0.00013632 | 1.367 | 1.41 |
| 14,15-EET | 0.00013835 | 1.287 | 1.41 |
| $PGD_2$ | 0.00097566 | $-1.214$ | 1.40 |
| 5-HETE | 0.00136022 | 1.489 | 1.29 |
| $PGF_{2\alpha}$ | 0.00151017 | $-1.296$ | 1.28 |
| tetranor-PGEM | 0.00053867 | 1.382 | 1.26 |
| 11,12-DHET | 0.00010749 | 1.504 | 1.24 |
| 20-HETE | 0.00014120 | 1.510 | 1.1 |

royalsocietypublishing.org/journal/rsob   Open Biol. 9: 180091

**Table 3.** Concentration of representative differential metabolites in human seminal plasma.

| | calculated concentration | |
|---|---|---|
| compound | healthy ($n = 33$) | asthenozoospermic ($n = 30$) |
| arachidonic acid (mg ml$^{-1}$) | $14.274 \pm 3.622$ | $17.170 \pm 4.073$ |
| 5-HETE (ng ml$^{-1}$) | $0.418 \pm 0.085$ | $0.607 \pm 0.123$ |
| 15-HETE (ng ml$^{-1}$) | $56.981 \pm 13.156$ | $80.018 \pm 16.048$ |
| 8,9-EET (ng ml$^{-1}$) | $48.587 \pm 8.082$ | $72.654 \pm 15.061$ |
| 14,15-DHET (ng ml$^{-1}$) | $0.061 \pm 0.012$ | $0.086 \pm 0.012$ |
| PGE$_2$ (mg ml$^{-1}$) | $10.041 \pm 2.284$ | $6.843 \pm 2.019$ |
| PGD$_2$ (mg ml$^{-1}$) | $12.262 \pm 2.251$ | $10.232 \pm 2.711$ |

## 3.2. Quality control and quantitative analysis of differential metabolites

The precision of QC samples was calculated by RSD of the peak area of IS. As shown in electronic supplementary material, table S3, the RSD value was within 15%, which qualified for biological sample analysis.

Seven out of 12 differential metabolites were selected for quantitative analysis: 5-HETE and 15-HETE (metabolized by LOX), 8,9-EET and 14,15-DHET (metabolized by CYP450), PGE$_2$ and PGD$_2$ (metabolized by COX) and AA. Within the validated concentration range (HETEs, EETs and DHETs: $0.01-50$ ng ml$^{-1}$, AA and PGs: $0.5-400$ ng ml$^{-1}$), the correlation coefficient of each analyte was higher than 0.99 with a weighting factor of the reciprocal of the concentration squared ($1/X^2$), exhibiting the good linearity of the method. Representative linear equations and correlation coefficients are summarized in electronic supplementary material, table S4. A satisfactory intra-day accuracy ranging from $-6.25\%$ to 6.98% of the relative error was obtained. Both intra-day and inter-day precisions were within 15% of the RSD. The results are given in electronic supplementary material, table S5. Quantitation results of seven differential metabolites are shown in table 3.

## 3.3. AA-reduced sperm motility via P38 MAPK

Sperm motility of the control group was 58.9%. AA (20, 40, 60 µM AA) decreased sperm motility to 55.8%, 52.8% and 44.6%, respectively, among which the 60 µM AA group changed significantly compared with the control group ($*p < 0.05$) (shown in figure 2a; individual data shown in electronic supplementary material, table S6). Therefore, 60 µM AA was used to establish the *in vitro* asthenozoospermia model in the following experiments.

Expressions of MAPK proteins were evaluated by western blot. As shown in figure 2b, the p-P38 level was obviously enhanced after AA administration, and the p-P38/P38 ratio showed a significant concentration-dependent increase after AA incubation. In detail, the p-P38/P38 ratios of the 20, 40 and 60 µM AA groups were 1.29-, 166- and 1.90-fold of those of the control group, respectively ($*p < 0.05$, $**p < 0.01$).

Levels of P38, p-ERK1/2, ERK1/2, p-JNK (46 and 54 kDa isoforms) and JNK remained unchanged, thus the ratios of p-ERK/ERK and p-JNK/JNK were also unchanged. This indicated that activation of P38 MAPK, rather than ERK1/2 or JNK, was involved in AA-induced sperm defects.

As seen in figure 2c, both AA and anisomycin significantly decreased sperm motility to 48.8% and 44.4% (a$*p < 0.05$), respectively, compared with the control group (57.8%). SB203580 preincubation recovered sperm motility to 55.8%, which was significantly different from the AA group (b$*p < 0.05$). No significant difference in sperm motility was observed between SB203580 and the control groups (individual data shown in electronic supplementary material, table S6). Western blot results (figure 2d) revealed that both AA and anisomycin significantly upregulated the p-P38/P38 ratio compared with the control group (a$*p < 0.05$, a$**p < 0.01$). SB203580 significantly reduced the AA-upregulated p-P38 level (b$*p < 0.05$, compared with the AA group), while no significant change was found between the control and SB203580 groups. This confirmed that AA regulated sperm motility via the P38 MAPK pathway.

## 3.4. Impact of AA metabolic pathways on MAPKs and sperm motility

Inhibitors of the AA metabolic pathways (COX, LOX and CYP450) were selected. Figure 2e shows that all these inhibitors significantly improved AA-reduced sperm motility to 52.2–56.6%, compared with the AA group (44.4%, $*p < 0.05$). Sperm motility of the inhibitor groups was not significantly different from that of the control group (58.1%, $p > 0.05$) (individual data shown in electronic supplementary material, table S6). In addition, inhibitor groups had no significant difference from each other.

AA-induced P38 activation could be abolished by these inhibitors, as shown in figure 2f. The p-P38/P38 ratio was increased after AA incubation, and was lower in the inhibitor groups than in the AA group. Similar to motility change, no significant differences were found between different inhibitors, as well as inhibitor groups and control group. This suggested that all three metabolic pathways of AA were involved in regulating P38 MAPK and further resulted in sperm defects. The intensities of the western blot results are shown in electronic supplementary material, table S7, and the original images of western blot results included in figure 2 are shown in electronic supplementary material.

Our *in vitro* study suggested that abnormal AA metabolic networks in seminal plasma could reduce sperm motility via P38 MAPK activation through LOX, CYP450 and COX metabolic pathways of AA.

royalsocietypublishing.org/journal/rsob Open Biol. 9: 180091

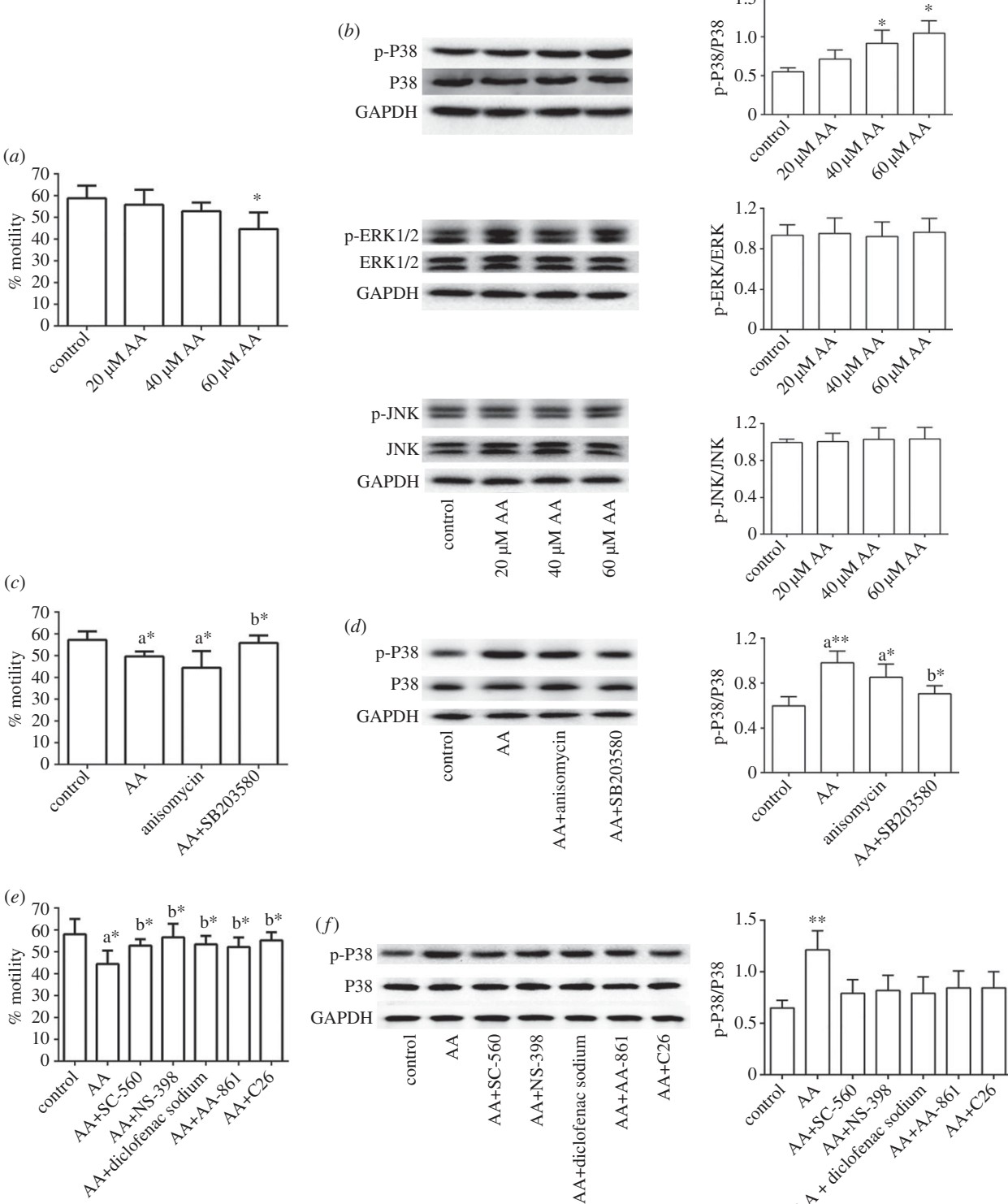

**Figure 2.** *In vitro* molecular mechanism study. (*a*) Percentages of motile spermatozoa after AA administration (*$p < 0.05$, compared with the control group). (*b*) Protein levels of MAPKs after AA treatment. Results are representative of three independent experiments (*$p < 0.05$, **$p < 0.01$, compared with the control group; Y-axis represents the intensity ratio of p-P38/P38, p-ERK/ERK and p-JNK/JNK, respectively, after adjustment by GAPDH). (*c*) Per cent motile spermatozoa after incubation of AA and specific MAPK inhibitors (a*$p < 0.05$, compared with the control group; b*$p < 0.05$, compared with the AA group). (*d*) AA-induced P38 activation was abolished by P38 inhibitor (a*$p < 0.05$, a**$p < 0.01$, compared with the control group; b*$p < 0.05$, compared with the AA group; Y-axis represents the intensity ratio of p-P38/P38 after adjustment by GAPDH). (*e*) Per cent motile spermatozoa after treatment with AA and AA metabolic pathway inhibitors (a*$p < 0.05$, compared with the control group; b*$p < 0.05$, compared with the AA group). F, AA-induced P38 activation was blocked by selective COX-1 inhibitor, selective COX-2 inhibitor, non-selective COX-1 and COX-2 inhibitor, 5-LOX inhibitor and CYP2J2 inhibitor (**$p < 0.01$, compared with the control group; Y-axis represents the intensity ratio of p-P38/P38 after adjustment by GAPDH).

## 4. Discussion

Current metabolomic analytical platforms mainly included NMR, GC-MS and LC-MS/MS, each of which has its own advantages and shortcomings. For instance, GC-MS behaved excellently for analysis of small molecular metabolites such as amino acids, glucose, nucleic acids, etc., while LC–MS/MS had high sensitivity for biomacromolecules like choline and

royalsocietypublishing.org/journal/rsob    Open Biol. 9: 180091

phospholipid such as lipids without complex sample preparation.

In the present study, we employed HPLC-ESI-MS/MS for metabolomic research of human seminal plasma to investigate AA metabolic alterations between asthenozoospermic volunteers and healthy ones. *In vitro* sperm experiments and western blot analysis were then performed for further mechanism exploration. The correlation among sperm motility, AA metabolic network and MAPK pathway was characterized by mutual verification of metabolomic and molecular mechanism studies.

Results from a targeted HPLC-ESI-MS/MS approach showed a significant disorder of AA metabolism in asthenozoospermic seminal plasma. An obvious separation of the healthy group from the asthenozoospermic group was observed in both the PCA and PLS-DA plots, reflecting a significant distinction of metabolites between the two groups.

Levels of AA and its metabolites metabolized by the LOX and CYP450 pathways were significantly higher than those in the healthy group, indicating the enhancement of LOX and CYP450 activity in asthenozoospermic seminal plasma. However, AA metabolites metabolized by COX exhibited relatively complicated alterations. $PGE_2$, $PGD_2$ and $PGF_{2\alpha}$ decreased, while tetranor-PGEM increased in asthenozoospermic volunteers compared with healthy subjects.

Many studies have reported that PGs, metabolites of AA, are closely associated with the MAPK pathway. MAPKs are key regulatory enzymes in cell signalling, participating in diverse cellular functions such as growth, differentiation, stress and apoptosis [35]. Almog *et al.* [26–28] reported that the MAPK signalling pathway was closely correlated with spermatogenesis, sperm meiosis, capacitation and acrosome reaction. Our *in vitro* experiments showed that exogenous AA treatment decreased sperm motility in a concentration-dependent manner, proving AA's reductive effect on sperm motility.

After AA administration, p-ERK1/2, ERK1/2, p-JNK, JNK and P38 remained unchanged, while p-P38 was upregulated in a concentration-dependent manner, indicating that activation of P38 MAPK by AA, instead of ERK1/2 or JNK, was correlated with AA-induced sperm defects. Use of P38 inhibitor SB203580 and P38 activator anisomycin confirmed that AA resulted in sperm motility descent via the P38 MAPK pathway. Both AA and anisomycin resulted in a marked decline in sperm motility, while SB203580 significantly improved AA-reduced sperm motility.

As we know, the AA metabolic network contains three pathways: COX, LOX and CYP450. According to our metabolomic study, LOX and CYP450 were activated in asthenozoospermic volunteers, while COX was irregularly disturbed. By *in vitro* experiments and western blot, we found inhibitors of LOX, CYP450 and COX could block P38 activation and improve AA-reduced sperm motility, indicating that all three metabolic pathways were involved in regulation of P38 MAPK and sperm motility, which was basically consistent with metabolomic analysis.

Consequently, an abnormal AA metabolic network in seminal plasma might reduce sperm motility via P38 MAPK activation through the LOX, CYP450 and COX metabolic pathways, which is a potential pathomechanism of asthenozoospermia.

## 5. Conclusion

In our research, HPLC-ESI-MS/MS was used for a metabolomics study of human seminal plasma, followed by *in vitro* experiments and western blot analysis for molecular mechanism study. Firstly, targeted metabolomics revealed an abnormal AA metabolic network in asthenozoospermic seminal plasma. AA and its main metabolites metabolized by three pathways in asthenozoospermic volunteers changed significantly compared with those in healthy subjects. Molecular mechanism study further confirmed that an abnormal AA metabolic network could reduce sperm motility via P38 MAPK activation through the LOX, CYP450 and COX metabolic pathways of AA.

Data accessibility. Figures and tables supporting this paper have been uploaded as electronic supplementary material.

Authors' contributions. L.Y., acquisition of data, data analysis and interpretation, manuscript writing; X.Y., acquisition of data, data analysis and interpretation, critical reading of manuscript; B.M., data analysis and interpretation, critical reading of manuscript; H.Y., data analysis and interpretation, critical reading of manuscript; X.S., acquisition of subjects, study concept, critical reading of manuscript; B.H., data analysis and interpretation, critical reading of manuscript; Q.Z., study concept and design, critical reading of manuscript.

Competing interests. We declare we have no competing interests.

Funding. This work was funded by the National Natural Science Foundation of China (grant nos. 81373478 and 81370750) and the Jiangsu Synergetic Innovation Center for Advanced Bio-Manufacture (grant no. XTD1819).

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
