## [Reviewer comments · Open Biology]

Review History

RSOB-18-0091.R0 (Original submission)

Review form: Reviewer 1

Recommendation

Major revision is needed (please make suggestions in comments)

Are each of the following suitable for general readers?

- a) **Title**
No
- b) **Summary**
Yes
- c) **Introduction**
Yes

Is the length of the paper justified?

Yes

Should the paper be seen by a specialist statistical reviewer?

Yes

Is it clear how to make all supporting data available?

No

Is the supplementary material necessary; and if so is it adequate and clear?

Yes

Do you have any ethical concerns with this paper?

No

Comments to the Author

In this manuscript Yu et al. study the correlation between the presence of arachidonic acid (AA) and its metabolites in human sperm with sperm reduced motility- asthenozoospermia. The authors suggest that AA higher content negatively affects sperm motility and that concentration of AA metabolites is significantly decreased in asthenozoospermic patients. The authors indicate that arachidonic acid content evaluation could be a valuable biomarker for fertility diagnostics. The experiments heavily rely on using mass spectrometry approach, which in its turn relies on the quality of the human sperm samples used. In this regard I have several major concerns:

1. Human sperm samples- have the fertility of human donors (control cohort) or their ability to undergo capacitation and hyperactivation been verified? If not, it should be stated.
2. What g-force was used to perform sperm centrifugation (so far only rpm is given)? Excessive g-force can damage human sperm and alter their composition and lipidomics. Usually, swim up is used to select the healthy and motile fraction of sperm.
3. How exactly sperm capacitation was performed? What is the composition of BWW medium used in these experiments? Did the medium contain sufficient amount of bicarbonate, and albumin to ensure capacitation? Was the incubation done at 37C with unrestricted access to CO₂ (such as 5% CO₂ incubator)- the "must-have" for capacitation? This part is not mentioned at all.
4. According to description, human sperm cells were capacitated for 1 hour- this is not enough time for human sperm cells, which require capacitation for at least 3 hours (better 4-5). One-hour capacitation is a good time for rodent sperm, but definitely not for the primate sperm.
5. How capacitation was confirmed? Tyrosine phosphorylation profile? Motility changes? Acrosome reaction status (FITC-PSA staining)? Unless the fact that sperm cells were indeed capacitated is proven, this reviewer has a hard time to believe that sperm were indeed capacitated.
6. AA could be also a metabolite of 2-arachidoyl-glycerol (2AG). Did authors look at the presence of 2AG and whether there is any correlation between 2AG presence and motility changes?
7. Figure 2E: it hard to see that data indicated by statistical differences of "*" are actually different, unless the individual data used to build the graph, and not only mean+/-sd/sem are given. It is a common trend now to show the individual data, as well as possible outliers, and not only "means".

Minor concern:

1. Not sure abbreviations such as MAPK is good to use in the title.

Review form: Reviewer 2

Recommendation

Major revision is needed (please make suggestions in comments)

Are each of the following suitable for general readers?

- a) **Title**
Yes
- b) **Summary**
Yes
- c) **Introduction**
Yes

Is the length of the paper justified?

Yes

Should the paper be seen by a specialist statistical reviewer?

Yes

Is it clear how to make all supporting data available?

Yes

Is the supplementary material necessary; and if so is it adequate and clear?

Yes

Do you have any ethical concerns with this paper?

No

Comments to the Author

The manuscript described an important finding that arachidonic acid, its lipoyxygenase and cytochrome P450 metabolites were abnormally increased in asthenozoospermic volunteers by means of metabolomics. Further studies revealed that decreased sperm motility and upregulation of sperm phosphor-P38 by arachidonic acid, which was confirmed by inhibition of its metabolic pathways. It is a good study in general and worth to be considered for publishing. However, scientific writing need to be improved before acceptance.

Fig. 2 All western blot images were manipulated and some were not aligned well, the original images are required to verify the results. How the intensity of the western blot bands was evaluated and the value should be presented for statistical analysis.

Supplement Fig 1 need to label what each peak stands for, otherwise difficult to understand

Decision letter (RSOB-18-0091.R0)

20-Jul-2018

Dear Professor Zhang,

We are writing to inform you that the Editor has reached a decision on your manuscript RSOB-18-0091 entitled "Abnormal arachidonic acid metabolic network may reduce sperm motility via P38 MAPK", submitted to Open Biology.

As you will see from the reviewers' comments below, there are a number of criticisms that prevent us from accepting your manuscript at this stage. The reviewers suggest, however, that a revised version could be acceptable, if you are able to address their concerns. If you think that you can deal satisfactorily with the reviewer's suggestions, we would be pleased to consider a revised manuscript.

The revision will be re-reviewed, where possible, by the original referees. As such, please submit the revised version of your manuscript within six weeks. If you do not think you will be able to meet this date please let us know immediately.

When submitting your revised manuscript, please respond to the comments made by the referee(s) and upload a file "Response to Referees" in "Section 6 - File Upload". You can use this to document any changes you make to the original manuscript. In order to expedite the processing of the revised manuscript, please be as specific as possible in your response to the referee(s).

Please see our detailed instructions for revision requirements
<https://royalsociety.org/journals/authors/author-guidelines/>

Sincerely,

The Open Biology Team
mailto: openbiology@royalsociety.org

Reviewer(s)' Comments to Author(s):

Referee: 1

Comments to the Author(s)

In this manuscript Yu et al. study the correlation between the presence of arachidonic acid (AA) and its metabolites in human sperm with sperm reduced motility- asthenozoospermia. The

authors suggest that AA higher content negatively affects sperm motility and that concentration of AA metabolites is significantly decreased in asthenozoospermic patients. The authors indicate that arachidonic acid content evaluation could be a valuable biomarker for fertility diagnostics. The experiments heavily rely on using mass spectrometry approach, which in its turn relies on the quality of the human sperm samples used. In this regard I have several major concerns:

1. Human sperm samples- have the fertility of human donors (control cohort) or their ability to undergo capacitation and hyperactivation been verified? If not, it should be stated.
2. What g-force was used to perform sperm centrifugation (so far only rpm is given)? Excessive g-force can damage human sperm and alter their composition and lipidomics. Usually, swim up is used to select the healthy and motile fraction of sperm.
3. How exactly sperm capacitation was performed? What is the composition of BWW medium used in these experiments? Did the medium contain sufficient amount of bicarbonate, and albumin to ensure capacitation? Was the incubation done at 37C with unrestricted access to CO₂ (such as 5% CO₂ incubator)- the "must-have" for capacitation? This part is not mentioned at all.
4. According to description, human sperm cells were capacitated for 1 hour- this is not enough time for human sperm cells, which require capacitation for at least 3 hours (better 4-5). One-hour capacitation is a good time for rodent sperm, but definitely not for the primate sperm.
5. How capacitation was confirmed? Tyrosine phosphorylation profile? Motility changes? Acrosome reaction status (FITC-PSA staining)? Unless the fact that sperm cells were indeed capacitated is proven, this reviewer has a hard time to believe that sperm were indeed capacitated.
6. AA could be also a metabolite of 2-arachidoyl-glycerol (2AG). Did authors look at the presence of 2AG and whether there is any correlation between 2AG presence and motility changes?
7. Figure 2E: it hard to see that data indicated by statistical differences of "*" are actually different, unless the individual data used to build the graph, and not only mean+/-sd/sem are given. It is a common trend now to show the individual data, as well as possible outliers, and not only "means".

Minor concern:

1. Not sure abbreviations such as MAPK is good to use in the title.

Referee: 2

Comments to the Author(s)

The manuscript described an important finding that arachidonic acid, its lipoxygenase and cytochrome P450 metabolites were abnormally increased in asthenozoospermic volunteers by means of metabolomics. Further studies revealed that decreased sperm motility and upregulation of sperm phosphor-P38 by arachidonic acid, which was confirmed by inhibition of its metabolic pathways. It is a good study in general and worth to be considered for publishing. However, scientific writing need to be improved before acceptance.

Fig. 2 All western blot images were manipulated and some were not aligned well, the original images are required to verify the results. How the intensity of the western blot bands was evaluated and the value should be presented for statistical analysis.

Supplement Fig 1 need to label what each peak stands for, otherwise difficult to understand

Author's Response to Decision Letter for (RSOB-18-0091.R0)

See Appendix A.

RSOB-18-0091.R1 (Revision)

Review form: Reviewer 2

Recommendation

Accept with minor revision (please list in comments)

Are each of the following suitable for general readers?

- a) **Title**
Yes
- b) **Summary**
Yes
- c) **Introduction**
Yes

Is the length of the paper justified?

Yes

Should the paper be seen by a specialist statistical reviewer?

No

Is it clear how to make all supporting data available?

Yes

Is the supplementary material necessary; and if so is it adequate and clear?

Yes

Do you have any ethical concerns with this paper?

No

Comments to the Author

The authors made a very good revision with most questions successfully rebutted. However, the requirement of uploading full gel Western Blot images is not fulfilled (still manipulated cut band). The authors specified that the band intensity was quantified using the ChemiScope 3400 Mini, the original intensity result (triplicates) should be listed in the supplementary and unit (if arbitrary) of Y-axis should be explained in Fig. 2 legend.

Decision letter (RSOB-18-0091.R1)

19-Sep-2018

Dear Professor Zhang

We are pleased to inform you that your manuscript RSOB-18-0091.R1 entitled "Abnormal arachidonic acid metabolic network may reduce sperm motility via P38 MAPK" has been accepted by the Editor for publication in *Open Biology*. The reviewer(s) have recommended publication, but also suggest some minor revisions to your manuscript. Therefore, we invite you to respond to the reviewer(s)' comments and revise your manuscript.

Please submit the revised version of your manuscript within 14 days. If you do not think you will be able to meet this date please let us know immediately and we can extend this deadline for you.

- 1) A text file of the manuscript (doc, txt, rtf or tex), including the references, tables (including captions) and figure captions. Please remove any tracked changes from the text before submission. PDF files are not an accepted format for the "Main Document".
- 2) A separate electronic file of each figure (tiff, EPS or print-quality PDF preferred). The format should be produced directly from original creation package, or original software format. Please note that PowerPoint files are not accepted.
- 3) Electronic supplementary material: this should be contained in a separate file from the main text and meet our ESM criteria (see <http://royalsocietypublishing.org/instructions-authors#question5>). All supplementary materials accompanying an accepted article will be treated as in their final form. They will be published alongside the paper on the journal website and posted on the online figshare repository. Files on figshare will be made available approximately one week before the accompanying article so that the supplementary material can be attributed a unique DOI.

Online supplementary material will also carry the title and description provided during submission, so please ensure these are accurate and informative. Note that the Royal Society will not edit or typeset supplementary material and it will be hosted as provided. Please ensure that

the supplementary material includes the paper details (authors, title, journal name, article DOI). Your article DOI will be 10.1098/rsob.2016[last 4 digits of e.g. 10.1098/rsob.20160049].

4) A media summary: a short non-technical summary (up to 100 words) of the key findings/importance of your manuscript. Please try to write in simple English, avoid jargon, explain the importance of the topic, outline the main implications and describe why this topic is newsworthy.

Images

Data-Sharing

It is a condition of publication that data supporting your paper are made available. Data should be made available either in the electronic supplementary material or through an appropriate repository. Details of how to access data should be included in your paper. Please see <http://royalsocietypublishing.org/site/authors/policy.xhtml#question6> for more details.

Data accessibility section

Sincerely,

The Open Biology Team
<mailto:openbiology@royalsociety.org>

Reviewer(s)' Comments to Author:

Referee: 2

Comments to the Author(s)

The authors made a very good revision with most questions successfully rebutted. However, the requirement of uploading full gel Western Blot images is not fulfilled (still manipulated cut band). The authors specified that the band intensity was quantified using the ChemiScope 3400 Mini, the original intensity result (triplicates) should be listed in the supplementary and unit (if arbitrary) of Y-axis should be explained in Fig. 2 legend.

Author's Response to Decision Letter for (RSOB-18-0091.R1)

See Appendices B & C.

Decision letter (RSOB-18-0091.R2)

05-Nov-2018

Dear Professor Zhang,

We are writing to inform you that the Editor has reached a decision on your manuscript RSOB-18-0091.R2 entitled "Abnormal arachidonic acid metabolic network may reduce sperm motility via P38 MAPK", submitted to Open Biology.

Thank you for taking the time to respond to the referees and Editors. The editors have discussed the feedback further and find that the Western blots in question in their current form prevent us from publishing the manuscript (please see the comments from the image audit attached).

In order to proceed, we would require additional/new Western blots documenting the same data set. If you are able to provide adequate Western blots for this data set, we would be pleased to accept the manuscript for publication.

When submitting your revised manuscript, please respond to the comments made by the referee(s) and upload a file "Response to Referees" in "Section 6 - File Upload". You can use this to document any changes you make to the original manuscript. In order to expedite the processing of the revised manuscript, please be as specific as possible in your response to the referee(s).

Please see our detailed instructions for revision requirements
<https://royalsociety.org/journals/authors/author-guidelines/>

Sincerely,

The Open Biology Team
mailto: openbiology@royalsociety.org

Decision letter (RSOB-18-0091.R3)

29-Mar-2019

Dear Professor Zhang

We are pleased to inform you that your manuscript entitled "Abnormal arachidonic acid metabolic network may reduce sperm motility via P38 MAPK" has been accepted by the Editor for publication in Open Biology.

Article processing charge

Please note that the article processing charge is immediately payable. A separate email will be sent out shortly to confirm the charge due. The preferred payment method is by credit card; however, other payment options are available.

Sincerely,

The Open Biology Team
mailto: openbiology@royalsociety.org

Appendix A

Dear Editor,

RE: Ms. No. RSOB-18-0091

We would like to thank you and the reviewers for the helpful comments and suggestions as well as the time and effort spent reviewing this manuscript. We have addressed all the reviewers' comments point by point and revised our manuscript accordingly. Please see below responses following each comment. Changes are either made via tracking tool or highlighted in red in the revised manuscript. We thank you for your editorial efforts in advance and look forward to the acceptance of this article.

Referee #1:

In this manuscript Yu et al. study the correlation between the presence of arachidonic acid (AA) and its metabolites in human sperm with sperm reduced motility- asthenozoospermia. The authors suggest that AA higher content negatively affects sperm motility and that concentration of AA metabolites is significantly decreased in asthenozoospermic patients. The authors indicate that arachidonic acid content evaluation could be a valuable biomarker for fertility diagnostics.

The experiments heavily rely on using mass spectrometry approach, which in its turn relies on the quality of the human sperm samples used. In this regard I have several major concerns.

Major concern:

1) Human sperm samples- have the fertility of human donors (control cohort) or their ability to undergo capacitation and hyperactivation been verified? If not, it should be stated.

Response: Thank you for your suggestion. The fertility and their ability to undergo capacitation and hyperactivation of human donors usually are not examined in outpatient department. Semen routine analysis includes other parameters such as semen volume, color, pH, vitality, motility, microscopy, etc. And we have deleted previous description of "capacitation" and modified relevant information in the experiment process. Thus fertility, capacitation and hyperactivation ability were not tested and not stated in this paper.

2) What g-force was used to perform sperm centrifugation (so far only rpm is given)? Excessive g-force can damage human sperm and alter their composition and lipidomics. Usually, swim up is used to select the healthy and motile fraction of sperm.

Response: Thank you for your kind suggestion. G-force (300 g) used to perform sperm centrifugation was now stated in Section 2.1.

In fact, swim up and Percoll gradient are both widely used to separate motile spermatozoa. Of course, only one of them could be used for one single experiment. It has been reported that the ability of Percoll gradient preparation to extract motile spermatozoa was higher than that of the swim up preparation [1, 2] and has been adopted in many literatures [3, 4]. Therefore, discontinuous Percoll gradient centrifugation was used in this paper.

3) How exactly sperm capacitation was performed? What is the composition of BWB medium used in these experiments? Did the medium contain sufficient amount of bicarbonate, and albumin to ensure capacitation? Was the incubation done at 37°C with unrestricted access to CO₂ (such as 5% CO₂ incubator)- the “must-have” for capacitation? This part is not mentioned at all.

Response: Thank you for your valuable comment. Spermatozoa were separated from semen by discontinuous Percoll gradient, as mentioned in the manuscript. Resuspended spermatozoa in BWB medium were incubated for 1 h at 37 °C in humidified air with 5% CO₂ in a cell culture incubator. This part has been added in Section 2.7 in the revised manuscript.

We checked the *in vitro* sperm experimental process in this study carefully. Actually, BWB incubation was a pre-incubation process, instead of sperm capacitation which could be confirmed, and was mis-defined as capacitation in the original manuscript.

Besides, sperm motility was what we studied on, rather than capacitation status. And in the previous literatures [5, 6], examination of sperm capacitation status was not involved in the studies which

focused on sperm motility. Thus capacitation was not necessary in our research. To avoid the confusion, we replaced previous description of “capacitation” with “preincubation”. And relevant information was modified in the revised manuscript.

In this study, BWW medium was a mature commercial product. Therefore, the composition was not described in the manuscript. It contains sodium chloride, potassium chloride, calcium chloride, potassium dihydrogen phosphate, magnesium sulfate, sodium bicarbonate, glucose, sodium pyruvate, sodium lactate, HEPES, bovine serum albumin.

4) According to description, human sperm cells were capacitated for 1 hour- this is not enough time for human sperm cells, which require capacitation for at least 3 hours (better 4-5). One-hour capacitation is a good time for rodent sperm, but definitely not for the primate sperm.

Response: Thank you for your valuable input. As mentioned in (3), capacitation was not performed in our research, so we have modified relevant description to avoid the confusion.

5) How capacitation was confirmed? Tyrosine phosphorylation profile? Motility changes? Acrosome reaction status (FITC-PSA staining)? Unless the fact that sperm cells were indeed capacitated is proven, this reviewer has a hard time to believe that sperm were indeed capacitated.

Response: Thank you for your comment. As mentioned in (3), capacitation was not performed or confirmed in our research, so we have modified relevant description to avoid the confusion.

6) AA could be also a metabolite of 2-arachidoyl-glycerol (2AG). Did authors look at the presence of 2AG and whether there is any correlation between 2AG presence and motility changes?

Response: Thank you for your valuable input. Our research was a targeted metabolomics research of the association between AA metabolic network and asthenozoospermia. What we focus on was AA

and its metabolites. Since AA could be a metabolite of 2-AG, we did not take 2-AG into consideration. The correlation between 2-AG presence and motility changes was not included in the study.

7) Figure 2E: it hard to see that data indicated by statistical differences of “*” are actually different, unless the individual data used to build the graph, and not only mean+/-sd/sem are given. It is a common trend now to show the individual data, as well as possible outliers, and not only “means”.

Response: Thank you for your valuable suggestion. Individual data of sperm motility have been uploaded as Supplement Material.

Minor concern:

1) Not sure abbreviations such as MAPK is good to use in the title.

Response: Thank you for your comment. MAPK is short for mitogen-activated protein kinase. Usually, when an abbreviation appears for the first time, the full name should be given. But mitogen-activated protein kinase is a little long to be included in the title. And MAPK is very familiar for researchers and is widely used in the title in many literatures [7-10]. Therefore the full name of MAPK has not been given in the title.

References

1. Englert Y., Vandenberg M., Rodesch C., Bertrand E., Biramane J., Legreve A. 1992 Comparative Auto-Controlled Study Between Swim-up And Percoll Preparation Of Fresh Semen Samples for Invitro Fertilization. *Hum Reprod* **7**(3), 399-402. (doi:DOI 10.1093/oxfordjournals.humrep.a137657).
2. Arias M.E., Andara K., Briones E., Felmer R. 2017 Bovine sperm separation by Swim-up and density gradients (Percoll and BoviPure): Effect on sperm quality, function and gene expression. *Reprod Biol* **17**(2), 126-132. (doi:10.1016/j.repbio.2017.03.002).
3. Paiva C., Amaral A., Rodriguez M., Canyellas N., Correig X., Balleca J.L., Ramalho-Santos J., Oliva R. 2015 Identification of endogenous metabolites in human sperm cells using proton nuclear magnetic resonance (H-1-NMR) spectroscopy and gas chromatography-mass spectrometry (GC-MS). *Andrology-Us* **3**(3), 496-505.

(doi:10.1111/andr.12027).

4. Aitken R.J., Ryan A.L., Curry B.J., Baker M.A. 2003 Multiple forms of redox activity in populations of human spermatozoa. *Mol Hum Reprod* **9**(11), 645-661. (doi:10.1093/molehr/gag086).
5. Francou M.M., Girela J.L., de Juan A., Ten J., Bernabeu R., De Juan J. 2017 Human sperm motility, capacitation and acrosome reaction are impaired by 2-arachidonoylglycerol endocannabinoid. *Histol Histopathol* **32**(12), 1351-1358. (doi:10.14670/HH-11-911).
6. Liu S.W., Li Y., Zou L.L., Guan Y.T., Peng S., Zheng L.X., Deng S.M., Zhu L.Y., Wang L.W., Chen L.X. 2017 Chloride channels are involved in sperm motility and are downregulated in spermatozoa from patients with asthenozoospermia. *Asian J Androl* **19**(4), 418-424. (doi:10.4103/1008-682X.181816).
7. Murphy L.O., Blenis J. 2006 MAPK signal specificity: the right place at the right time. *Trends Biochem Sci* **31**(5), 268-275. (doi:10.1016/j.tibs.2006.03.009).
8. Liu Z.Y., Wang B., He R.J., Zhao Y.M., Miao L. 2014 Calcium signaling and the MAPK cascade are required for sperm activation in *Caenorhabditis elegans*. *Bba-Mol Cell Res* **1843**(2), 299-308. (doi:10.1016/j.bbamcr.2013.11.001).
9. Qi J., Xian X.H., Li L., Zhang M., Hu Y.Y., Zhang J.G., Li W.B. 2018 Sulbactam Protects Hippocampal Neurons Against Oxygen-Glucose Deprivation by Up-Regulating Astrocytic GLT-1 via p38 MAPK Signal Pathway. *Front Mol Neurosci* **11**. (doi:Artn 28110.3389/Fnmol.2018.00281).
10. Qi F.F., Bai S., Wang D.D., Xu L., Hu H.Y., Zeng S., Chai R.N., Liu B.X. 2017 Macrophages produce IL-33 by activating MAPK signaling pathway during RSV infection. *Mol Immunol* **87**, 284-292. (doi:10.1016/j.molimm.2017.05.008).

Referee #2:

The manuscript described an important finding that arachidonic acid, its lipoxygenase and cytochrome P450 metabolites were abnormally increased in asthenozoospermic volunteers by means of metabolomics. Further studies revealed that decreased sperm motility and upregulation of sperm phosphor-P38 by arachidonic acid, which was confirmed by inhibition of its metabolic pathways. It is a good study in general and worth to be considered for publishing.

1) Scientific writing need to be improved before acceptance.

Response: Thank you for your suggestion. We have made corrections to those spelling and grammatical errors. And the paper was extensively modified to improve scientific writing. Please let us know if another round of revision is needed.

2) Fig. 2 All western blot images were manipulated and some were not aligned well, the original images are required to verify the results. How the intensity of the western blot bands was evaluated and the value should be presented for statistical analysis.

Response: Thanks for your advice. Original western blot images have been uploaded as Supplement Material to verify the results. Protein bands were visualized by enhanced chemiluminescence (ECL, Thermo Scientific, USA) and the intensity was quantified using the ChemiScope 3400 Mini (Clinx Science Instruments, China). The relevant information has been added in Section 2.10 in the revised manuscript.

3) Supplement Fig 1 need to label what each peak stands for, otherwise difficult to understand

Response: Thanks for your valuable advice. Each peak in Supplement Fig 1 has been labelled now. Besides, we reordered the compounds in Table 1 which listed the AA metabolites detected in human seminal plasma according to the retention time of the metabolites to help understanding.

Best regards,
Dr. Qi Zhang

Appendix B

Dear Editor,

RE: Ms. No. RSOB-18-0091.R1

We would like to thank you and the reviewers so much for the helpful comments and valuable advices as well as the time and effort spent reviewing our manuscript entitled "Abnormal arachidonic acid metabolic network may reduce sperm motility via P38 MAPK". We have addressed the Referee 2's comments point by point and revised our manuscript accordingly. Please see below responses following each comment. Changes are highlighted in red in the revised manuscript. We thank you so much for your editorial efforts.

Referee #2:

The authors made a very good revision with most questions successfully rebutted.

1) However, the requirement of uploading full gel Western Blot images is not fulfilled (still manipulated cut band).

Response: Thank you for your comment. Actually, the individual western blot images we uploaded were the original images.

In our WB experiment, the protein samples are separated by vertical electrophoresis on 10% SDS-PAGE (Fig.a). The approximate position of target protein band is determined by prestained protein marker (Fig.b). The gel was then cut and transferred onto a PVDF membrane (Fig.c), which was then blocked by 5% nonfat milk for 1 h and react with first antibodies at 4 °C overnight. Horseradish peroxidase-conjugated secondary antibodies were subsequently used to incubate and visualize the protein bands by chemiluminescence method. Horseradish enzyme catalyzed the chemiluminescence reaction of ECL solution A and B, and the intensity was detected by ChemiScope 3400 Mini instrument (Fig.d). After the protein was transferred from the gel onto the PVDF membrane,

all subsequent operations were performed with the PVDF membrane, so the gel was discarded after the protein was transferred (Fig.e). Unlike DNA Agarose Gel Electrophoresis (Fig.f) whose results are analyzed by gel imaging system, our WB results are analyzed by the chemiluminescence of PVDF membrane, therefore the full gel of Western Blot images are not given.

Fig.a SDS-PAGE vertical electrophoresis.

Fig.b Protein sample and marker on gel separated by SDS-PAGE.

Fig.c Transfer unit.

Fig.d Chemiluminescence system (ChemiScope 3400 Mini).

Fig.e PVDF membrane and gel after transfer.

Fig.f Gel imaging system

2) The authors specified that the band intensity was quantified using the ChemiScope 3400 Mini, the original intensity result (triplicates) should be listed in the supplementary and unit (if arbitrary) of Y-axis should be explained in Fig. 2 legend.

Response: Thanks for your valuable advice. The original intensity results have been added in supplementary material. And Y-axis has now been explained in Fig. 2 legend.

Sincerely,

Dr. Qi Zhang

Mail to: nancyzhang03@hotmail.com

Appendix C

Dear Editor,

RE: Ms. No. RSOB-18-0091.R2

We would like to thank you and the reviewers so much for the helpful comments and valuable advices as well as the time and effort spent reviewing our manuscript entitled "Abnormal arachidonic acid metabolic network may reduce sperm motility via P38 MAPK". We thank you so much for your editorial efforts.

1) We need you to explain the reason for the anomalies in the background of the beta actin panels in Figures 2B and D. We have attached our image auditors report. Please respond to it by uploading your response in the 'response to referees'.

Image Report:

- a. The p-P38 and the P38 blots look very similar in shape. The same applies to the p-ERK1/2 and the ERK1/2 panels. The bands are almost identical in shape. Is this feasible?
- b. The β -actin panel looks manipulated, the background appears to have been erased. The areas are marked by red arrows. Uncharacteristically sharp edges in the background are circled.

Response: Thank you for your valuable input.

- a. In Figure 2B, the p-P38 and the P38, as well as the ERK and p-ERK, actually were the same in shape. Because after protein bands of p-P38 were visualized by chemiluminescence method, the first antibodies and secondary antibodies were removed from PVDF membrane by using "stripping buffer". The regenerated PVDF membrane was then blocked by 5% nonfat milk for 1 h and react with P38 first antibody, respectively, at 4 °C overnight. Horseradish peroxidase-conjugated secondary antibodies were subsequently used to incubate and visualize the protein bands by chemiluminescence method. The same applies to the p-ERK1/2 and the ERK1/2 bands. Therefore, the protein bands of p-P38

and the P38, as well as ERK and p-ERK, were the same in shape.

b. The β -actin panel looks manipulated. The gaps between P38 bands were wider than those between original actin bands, because different sizes of swim lanes were made. For the sake of intuition and visualization, actin panel were manipulated, as shown in the figure below. If the β -actin bands in Figure 2 B and D need to be re-manipulated, please let us know.

Please find original WB images without manipulation in Supplementary Material.

2) You must clarify whether you no longer have the original uncut blots available for inspection?

Response: The original protein bands were uploaded as **Supplementary Data** (Original images of western blot results included in Fig 2.) which can be used for inspection.

As described in the manuscript, in our WB experiments, the protein samples are separated by vertical electrophoresis on 10% SDS-PAGE. The approximate position of target protein band is determined by pre-stained protein marker. The gel was then **cut** and transferred onto a PVDF membrane, which was then blocked by 5% nonfat milk for 1 h and react with first antibodies at 4 °C overnight. Horseradish peroxidase-conjugated secondary antibodies were subsequently used to incubate

and visualize the protein bands by chemiluminescence method.

After the protein band was cut and transferred from the gel onto the PVDF membrane, all subsequent operations were performed with the PVDF membrane, so the PVDF membrane was cut to the right size with target protein band, and the gel was discarded after the protein was transferred. Therefore **Supplementary Data** we uploaded were the original WB images.

Please find detailed information and images of WB procedure in **Response to Referees**.

Sincerely,

Dr. Qi Zhang

Mail to: nancyzhang03@hotmail.com